# Automotive-ENV: Benchmarking Multimodal Agents in Vehicle Interface Systems

## Abstract

Multimodal agents have demonstrated strong performance in general GUI inter-
actions, but their application in automotive systems has been largely unexplored.
In-vehicle GUIs present distinct challenges: drivers' limited attention, strict safety
requirements, and complex location-based interaction patterns. To address these
challenges, we introduce Automotive-ENV, the first high-fidelity benchmark and
interaction environment tailored for vehicle GUIs. This platform defines 185 pa-
rameterized tasks spanning explicit control, implicit intent understanding, and
safety-aware tasks, and provides structured multimodal observations with precise
programmatic checks for reproducible evaluation. Building on this benchmark, we
propose ASURADA, a geo-aware multimodal agent that integrates GPS-informed
context to dynamically adjust actions based on location, environmental conditions,
and regional driving norms. Experiments show that geo-aware information signif-
icantly improves success on safety-aware tasks, highlighting the importance of
location-based context in automotive environments. We will release Automotive-
ENV, complete with all tasks and benchmarking tools, to further the development
of safe and adaptive in-vehicle agents.

## 1 Introduction

Autonomous agents that interpret natural language instructions and control graphical user inter-
faces (GUI) can provide enormous value to users by automating repetitive tasks, augmenting hu-
man cognitive capabilities, and accomplishing complex workflows (Gravitas, 2023; Wu et al., 2023;
Xie et al., 2023; Yao et al., 2022b; Yang et al., 2023b; Ding, 2024; Park et al., 2023). To real-
ize this potential, current research efforts have primarily focused on building and evaluating GUI
agents capable of operating within desktop operating systems, mobile applications, and web envi-
ronments (Deng et al., 2023; Rawles et al., 2023; Zheng et al., 2024a; Koh et al., 2024; Kim et al.,
2024; He et al., 2024), establishing important foundations for GUI automation research. These
existing evaluation methods typically rely on static interface screenshots and user instructions as
input, measuring performance by comparing agent behaviors with pre-collected human demonstra-
tions (Deng et al., 2023; Rawles et al., 2023; Toyama et al., 2021; Li et al., 2024; Chai et al., 2024;
Xie et al., 2024; Baek & Bae, 2016). Such approaches work well in traditional computing envi-
ronments because desktop and mobile devices operate in relatively stable and controlled scenarios
where device state has limited impact on task execution. However, this focus represents only a sub-
set of the diverse interface ecosystems that people interact with daily, notably excluding In-vehicle
GUI systems that support navigation, communication, media, and safety functions in millions of
automobiles worldwide.

In-vehicle GUI systems introduce evaluation challenges that existing methods cannot adequately
address. First, automotive agents operate in highly dynamic and safety-critical contexts, where
factors such as real-time location, driving state, weather, and traffic conditions directly determine
correct task execution (Zhou et al., 2023; Koh et al., 2024). For example, as shown in Figure 1, the
seemingly simple command "I can't see through the windshield, it's all fogged up" requires the agent
to first perform contextual reasoning over current driving conditions, and then correctly operate the
interface (e.g., enabling the front defroster). Second, because drivers must prioritize road attention,
their commands are typically brief, ambiguous, or incomplete, forcing agents to infer intent from
limited information. Third, mistakes in automotive tasks can have immediate safety implications:
a single incorrect navigation instruction or inappropriate system response may distract the driver or

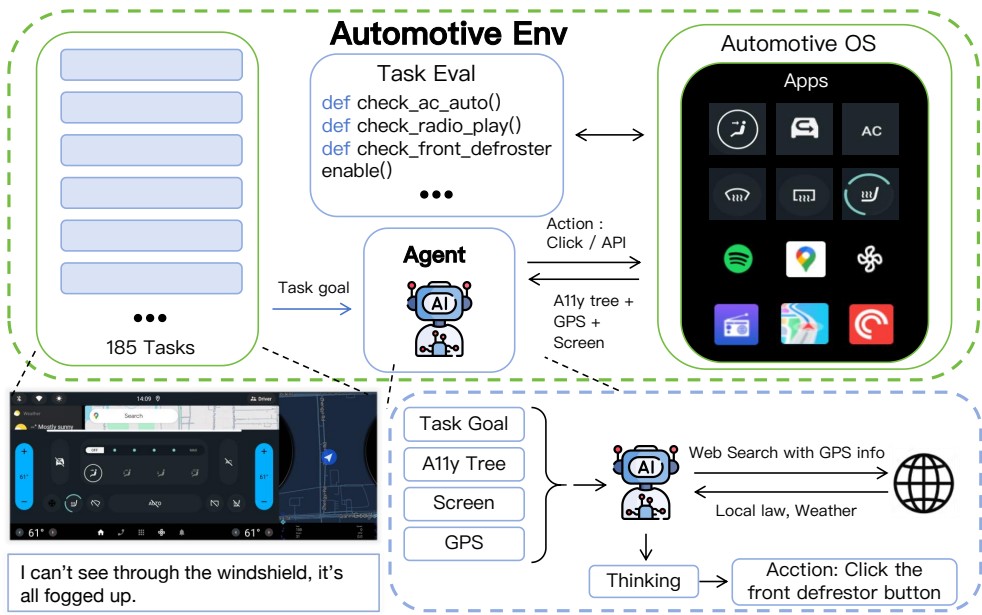

Figure 1: Automotive OS-based environment where the agent observes the accessibility tree, screen, and GPS; optionally consults GPS-contextualized web knowledge; and acts through tap screens and API calls. Task success is determined by low-level programmatic checks of the UI state and system signals.

induce hazardous behavior. Existing evaluation frameworks, centered on interface screenshots and static inputs, fail to capture these challenges as they lack awareness of vehicle state, environmental conditions, and safety constraints, and cannot assess an agent's adaptability or reliability under real-time driving dynamics (Liu et al., 2023; Wu et al., 2024).

To address these challenges, we introduce Automotive-ENV, a comprehensive evaluation platform built on a real in-car operating system spanning 8 functional modules with 185 parameterized tasks. Unlike prior benchmarks based on synthetic interfaces or static specifications, Automotive-ENV dynamically instantiates tasks with randomly generated parameters, creating millions of unique scenarios that require agents to generalize across diverse interface states and driving contexts. Our platform leverages production-grade automotive software architectures and their embedded event-handling mechanisms to ensure robust reward signal generation under the safety-aware conditions characteristic of real automotive environments. Beyond the core automotive tasks, we extend Automotive-ENV by integrating external geographic, environmental, and sensor-driven scenarios, thereby enriching the diversity of evaluation conditions and enabling comprehensive assessment across varied driving contexts. Meanwhile, this platform is designed for practical deployment and broad accessibility, requiring less than 4 GB of memory and 10 GB of disk space while connecting agents to automotive systems through standard APIs without proprietary hardware requirements.

To demonstrate the utility of Automotive-ENV, we develop ASURADA (Automotive Multimodal Agent), a prototype multimodal agent designed to address the unique challenges of in-vehicle GUI environments. Unlike desktop or mobile GUIs, automotive tasks are inherently geo-dependent: user needs and system behaviors vary significantly with GPS location, traffic conditions, and regional driving rules. For example, the seemingly simple utterance "Adjust the air conditioning temperature" may require different actions depending on whether the vehicle is driving through a hot coastal city, a cold mountainous region, or a humid rainy environment. Motivated by this, ASURADA incorporates a novel GPS-informed context integration that conducts reasoning over GPS signals to infer environmental context and location-specific driving regulations. We evaluate ASURADA under both GPS-enhanced multimodal input—screenshot, text, and GPS—and GPS-absent input with only screenshots and text, across realistic scenarios ranging from congestion rerouting to climate control adjustments. Results show that while incorporating geographic context enhances robustness in safety-aware tasks, substantial challenges remain: ASURADA achieves a 65% success rate, outperforming adapted web-based GUI agent baselines but still falling far below human performance at

| Dataset | Env? | # Apps/Web | # Templates | Instances | Reward Method | Platform |
|---|---|---|---|---|---|---|
| GAIA | No | n/a | 466 | 1 | text-match | None |
| Mind2Web | No | 137 | 2350 | 1 | None | Desktop Web |
| WebLINX | No | 155 | 2337 | 1 | None | Desktop Web |
| WebVoyager | No | 15 | 643 | 1 | LLM judge | Desktop Web |
| PixelHelp | No | 4 | 187 | 1 | None | Android |
| MetaGUI | No | 6 | 1125 | 1 | None | Android |
| MoTiF | No | 125 | 4707 | 1 | None | Android (Apps+Web) |
| AitW | No | 357+ | 30378 | 1 | None | Android (Apps+Web) |
| AndroidControl | No | 833 | 15283 | 1 | None | Android (Apps+Web) |
| OmniAct | No | 60+ | 9802 | 1 | None | Desktop (Apps+Web) |
| AndroidArena | No | 13 | 221 | 1 | Action match / LLM | Android (Apps+Web) |
| LLamaTouch | No | 57 | 496 | 1 | Screen match | Android (Apps+Web) |
| MiniWoB++ | Yes | 1 | 114 | - | HTML/JS state | Web (synthetic) |
| WebShop | Yes | 1 | 12000 | 1 | product attr match | Desktop Web |
| WebArena | Yes | 6 | 241 | 3.3 | URL/Text match | Desktop Web |
| VisualWebArena | Yes | 4 | 314 | 2.9 | URL/Text/Image match | Desktop Web |
| WorkArena | Yes | 1 | 29 | 622.4 | cloud state | Desktop Web |
| Mobile-Env | Yes | 1 | 13 | 11.5 | regex | Android (Apps) |
| B-MoCA | Yes | 4 | 6 | 1.9 | regex | Android (Apps+Web) |
| MMInA | Yes | 14 | 1050 | 1 | text-match | Desktop Web |
| OSWorld | Yes | 9 | 369 | 1 | device/cloud state | Desktop (Apps+Web) |
| WindowsAgentArena | Yes | 11 | 154 | 1 | device state | Desktop (Apps+Web) |
| AgentStudio | Yes | 9 | 205 | 1 | device state | Desktop (Apps+Web) |
| AndroidWorld | Yes | 20 | 116 | $\infty$ | device state | Android (Apps+Web) |
| **Automotive-ENV** | Yes | 8 | 185 | $\infty$ | device state | Automotive OS |

Table 1: Comparison of different datasets and environments for benchmarking computer agents.

100%, underscoring both the necessity of geo-aware reasoning and the current limitations of reliable automotive GUI automation.

In summary, our main contributions are as follows:

- We introduce Automotive-ENV, a high-fidelity evaluation platform for in-vehicle GUI systems that balances generality and safety. It supports multimodal interactions, structured observations, and programmatic feedback to comprehensively assess agent robustness and generalization.

- We develop ASURADA, a structured VLM-based agent architecture that integrates perception, intent understanding, planning, and execution. A GPS reasoning module is incorporated to adapt agent behavior to geographic context and regional driving rules, improving robustness across diverse driving environments.

- We demonstrate that agents can leverage GPS to perceive richer environmental context and support decision-making, leading to significant improvements in reliability and responsiveness on safety-critical tasks.

## 2 RELATED WORK

### 2.1 DYNAMIC AGENT EVALUATION PLATFORMS

Building reliable autonomous agents necessitates evaluation frameworks that simulate authentic interaction scenarios while delivering precise feedback mechanisms for task assessment (Rawles et al., 2023; Deng et al., 2023; Abramson et al., 2022; Ruan et al., 2023; Chen et al., 2021). Current evaluation platforms predominantly focus on web navigation and general computing tasks. For instance, MiniWoB++ (Shi et al., 2017; Liu et al., 2018) offers compact synthetic HTML environments with configurable task parameters, while WebShop (Yao et al., 2023) creates simulated online retail scenarios. More comprehensive platforms like WebArena (Zhou et al., 2023) and VisualWebArena (Koh et al., 2024) encompass multi-domain website simulations. In the desktop computing space, platforms such as OSWorld (Xie et al., 2024), WindowsAgentArena (Bonatti et al., 2024), and AgentStudio (Zheng et al., 2024b) deliver comprehensive testing frameworks spanning 9-11 applications. Mobile agent evaluation has been addressed through B-MoCA (Lee et al., 2024), which examines 6 fundamental tasks across 4 applications, and Mobile-Env (Zhang et al., 2024), providing

13 task configurations within a single application environment. Table 1 compares existing evaluation environments for autonomous UI agents, but none address automotive-specific requirements such as constrained driver attention, safety-first design principles, and context-dependent task prioritization, underscoring the need for a dedicated framework for in-vehicle GUI systems

## 2.2 Autonomous Language Agents

Recent advances have demonstrated the remarkable potential of *language agents*—sophisticated language models designed to interact with external environments and other agents for complex task solving (Li et al., 2023; Wu et al., 2024). Current approaches predominantly fall into two categories: inference-based systems that leverage large language models (LLMs) such as GPT-4 for reasoning and planning through carefully designed prompt engineering (Shen et al., 2023; Yan et al., 2023), and trainable, open-source alternatives that prioritize customization flexibility and privacy preservation (Shao et al., 2023). While GPT-based agents like AutoGPT and HuggingGPT demonstrate impressive generalization capabilities across diverse domains, they suffer from limited adaptability when deployed in specialized environments with unique constraints and requirements. To address this limitation, the research community has increasingly focused on trainable methodologies that enable environment-specific optimization. Notable examples include m-BASH (Sun et al., 2022), which introduced ROI pooling techniques for GUI interaction tasks, Auto-UI (Zhang & Zhang, 2023), which reformulated GUI interactions as visual question answering problems, and CogAgent (Hong et al., 2023), which incorporated high-resolution visual processing modules with specialized alignment pretraining. However, existing GUI agents are primarily designed for desktop and mobile environments, failing to adequately address the unique challenges of automotive contexts, such as driver attention constraints, safety-critical interaction requirements, and the need for dynamic behavioral adaptation based on geographic location.

# 3 Automotive Environment

## 3.1 Automotive OS as an Agent Environment

As shown in Firgure 2, Automotive OS provides an ideal environment for developing autonomous agents in intelligent vehicles. Widely adopted in modern electric and premium vehicles, it delivers a unified software architecture for managing essential cockpit functions, including climate control, seat adjustment, wipers, multimedia systems, safety alerts, and energy management. Unlike traditional vehicle,s where displays served primarily informational purposes, Automotive OS GUIs have evolved into central interaction hubs that coordinate both user input and system-level operations.

A key advantage lies in deployment simplicity: the entire cockpit can be virtually simulated on standard laptop hardware without requiring specialized equipment. The platform supports modular configuration, state injection, and GUI playback capabilities, making it both practical and reproducible for multimodal agent research.

Compared to desktop or mobile operating systems, Automotive OS presents distinct challenges for agent development. While GUI are often simplified due to safety requirements, the action space becomes more constrained and highly context-dependent. Agent behavior must adapt to driving conditions, environmental factors (speed, weather), and user preferences. Successful agents must seamlessly integrate multimodal inputs—instruction commands, touch interactions, and sensor data—with precise API-driven control systems, delivering responses that are timely, interpretable, and safety-compliant. These requirements necessitate a new generation of GUI agents that demonstrate context awareness, safety sensitivity, and robust generalization across diverse automotive scenarios.

## 3.2 Observation and Action Space

The system provides a comprehensive interface enabling agents to receive observations and execute actions within automotive GUI platforms through standardized middleware frameworks and communication protocols.

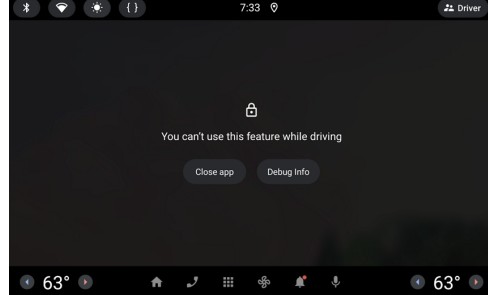

Figure 2: GUI cases of Automotive OS. The left image shows general in-car functions such as climate control and media playback, while the right image displays a safety warning interface triggered when the system detects unsafe driving behavior or hazardous driving conditions.

**Observation Space:** Agents access display captures, real-time vehicle state information, structured UI representations, GPS location data, and network connectivity status. This multi-layered approach allows agents to perceive both immediate environmental context and external information sources.

**Action Space:** Agents primarily interact through GUI interfaces, supporting classic touchscreen operations including tapping, swiping, and text input. Beyond these basic GUI interactions, the platform exposes automotive-specific safety-related APIs, such as emergency alert message pop-ups, enabling agents to execute critical safety function calls to ensure driving safety.

### 3.3 Reproducibility Framework

To ensure consistent evaluation under realistic conditions, Automotive-ENV implements strict control mechanisms over vehicle and system states. All tasks execute within a fixed simulation environment that accurately represents modern vehicle GUI architectures, utilizing a consistent software image based on an emulated Automotive OS.

**State Management** To guarantee consistency and reproducibility, the evaluation environment is designed with three complementary principles: state management, offline operation, and structured task execution. Before each task, the system time resets to predetermined values, ensuring consistent time-sensitive behavior, while application versions remain fixed—open-source components are sourced from verified repositories and Original Equipment Manufacturer system applications are preserved within the static vehicle image. All tasks run fully offline without login requirements or cloud dependencies, with generated data stored locally to maintain identical system states across runs. Each evaluation further incorporates explicit initialization routines, reward computation, and cleanup procedures, together enabling reliable and repeatable experimentation.

**Geographic Parameterization** Beyond static configurations, Automotive-ENV incorporates a sophisticated geographically-aware task parameterization system. This mechanism dynamically generates task parameters based on either the agent's current GPS location or predefined regional contexts (local climate patterns, traffic regulations, cultural user behavior norms) while maintaining valid and consistent evaluation criteria.

**Rewards Signal** Automotive-ENV provides stable and high-precision reward signals by directly accessing low-level system states through the native APIs of Automotive OS. Unlike approaches that rely on traditional vehicle communication protocols such as Controller Area Network (CAN) (Etschberger et al., 2001) bus or On-Board Diagnostics (OBD-II) (Michailidis et al., 2025), Automotive-ENV leverages operating system–level interfaces to query the internal status of key subsystems, including climate control, media playback, navigation, and network connectivity, as shown in Table 2. This allows agents to accurately determine task completion based on system feedback—for example, verifying whether the temperature has been set to the target value, whether navigation has successfully started to the specified destination, or whether the media player has switched to the requested content.

Compared to UI-based validation, system-state-based reward mechanisms are significantly more robust and platform-agnostic, avoiding misjudgment caused by visual differences across user inter-

| User Instruction (Natural Language) | Validation Logic |
|---|---|
| [Explicit Control] Turn the fan speed to Max. | `check_fan_speed_max()` |
| [Explicit Control] Turn on driver seat heater. | `check_driver_seat_heater_enable()` |
| [Implicit Intent] My hands are freezing. | `check_ac_auto()` |
| [Implicit Intent] Feels a bit lonely driving in silence. | `check_media_play()` |
| [Driving Alignment] The front window is foggy. | `check_front_defroster_enable()` |
| [Driving Alignment] I can't see through the windshield; it's all fogged up. | `check_front_defroster_enable()` |
| [Environment Alerts.] The rear window is fogging up too. | `check_raw_defroster_enable()` |
| [Environment Alerts.] I can barely see anything on this dark screen. | `check_screen_brightness()` |

Table 2: Representative user instructions for in-vehicle tasks, categorized by task type, with corresponding validation methods.

faces. Moreover, many system modules are shared across different vehicle applications—for instance, various infotainment systems often interface with the same HVAC controller—enabling high reusability of validation logic across tasks. This design provides a solid foundation for large-scale task definition, fine-grained agent evaluation, and reinforcement learning–based training.

### 3.4 TASK TAXONOMY

As shown in Figure 3, to systematically evaluate agent capabilities in real-world in-vehicle environments, we categorize the tasks in Automotive-ENV into two major types: *General Tasks* and *Safety-Aware Tasks*. This taxonomy covers both the functional requirements of everyday in-car interactions and the safety-critical aspects of real driving contexts.

**General Tasks**    General tasks focus on routine in-vehicle interactions, emphasizing functional correctness, natural interaction, and execution efficiency.

- Explicit Control: Tasks where the user issues clear and direct commands that can be mapped to GUI actions or backend APIs. For example, "Set the temperature to 22 degrees" can be directly translated into a call to the heating, ventilation, and air conditioning (HVAC) system.
- Implicit Intent: Tasks where user needs are expressed indirectly and ambiguously, requiring the agent to perform reasoning over linguistic and contextual cues. For example, the utterance "It feels stuffy in here" does not contain an explicit action command, yet the agent should infer that the intended operation is to improve ventilation or open a window.

**Safety-Aware Tasks**    Safety-aware tasks emphasize adaptation to driving states and environmental conditions, ensuring that agent behavior aligns with local regulations and safety requirements.

- Driving Alignment: The agent must adjust its driving behavior to comply with regional regulations and cultural norms. For example, automatically switching off high beams when driving on roads where their use is prohibited.
- Environment Alerts: The agent must continuously monitor in-vehicle and external conditions, issuing alerts or taking proactive actions to reduce risk. For instance, enabling fog lights in low-visibility weather or adjusting cabin temperature during extreme heat.

This twofold categorization enables Automotive-ENV to evaluate agents across both functional and safety dimensions—assessing not only their ability to reliably execute everyday user commands but also their capacity to make context-sensitive decisions that ensure safe and adaptive driving. In addition, all task instructions were manually validated to ensure stability and executability, while low-quality or ambiguous tasks were discarded, thereby guaranteeing the validity and fairness of the evaluation process.

## 4 AUTOMOTIVE CONTROL AGENT

Traditional in-vehicle GUI agents typically operate in isolation, relying only on screen observations and internal vehicle states. This limits their adaptability to dynamic driving environments and

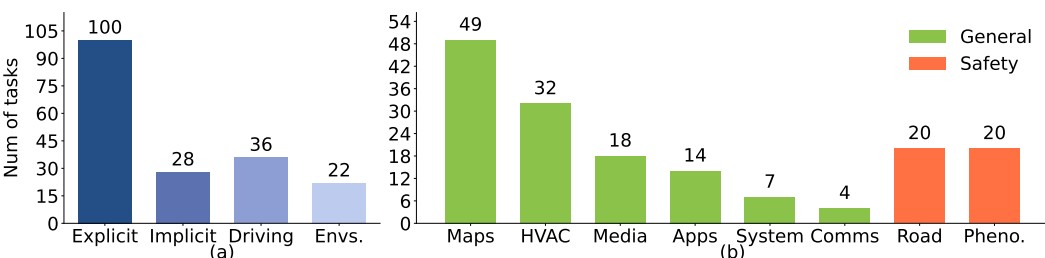

Figure 3: Task distributions across different dimensions. (a) Distribution of tasks by task dimensions. (b) Distribution of tasks across task categories (Maps, HVAC, Road, Phenomenon, Media, Apps, System, Comms).

diverse cultural contexts. To overcome these limitations, we propose **ASURADA**, a geo-adaptive multimodal agent for automotive systems.

ASURADA introduces two key innovations. First, it integrates real-time GPS information as an additional input modality, alongside screen content and accessibility tree elements. Second, it functions as a virtual sensor by issuing network-based queries informed by GPS data to retrieve external context such as weather, traffic regulations, safety requirements, and cultural driving norms. This enables context-aware decision-making beyond the vehicle's internal signals.

The agent's workflow proceeds in iterative cycles: it collects multimodal observations (vehicle state, GUI layout, accessibility tree, historical memory, GPS signals), generates structured JSON action plans with reasoning traces, executes actions through GUI or vehicle-level APIs, reflects on pre- and post-action differences, and updates memory to reinforce effective strategies. This design allows ASURADA to adapt its behavior to geographic and cultural contexts. For example, when handling the instruction "I feel hot," the agent may suggest opening windows in cool regions but directly activate air conditioning in persistently hot climates. Similarly, GPS-based regulation checks ensure automatic compliance with local speed limits and equipment requirements. Through this integrated cycle, ASURADA progressively develops context-sensitive, culturally aware, and safety-critical control behaviors, improving robustness in real-world automotive environments.

## 5 EXPERIMENTS

### 5.1 SETUP

We evaluate our approach on Automotive-ENV, including two task categories: (i) General Tasks, covering explicit control and implicit intent, and (ii) Safety-Aware Tasks, covering driving alignment and environment alerts. Task success rate is used as the evaluation metric, with human annotators serving as an upper bound.

We compare three agent variants: (1) T3A (Rawles et al., 2024), a text-only baseline that relies on user instructions and accessibility tree elements; (2) M3A (Rawles et al., 2024), a multimodal agent adapted from Android GUI control, which employs ReAct-style prompting (Yao et al., 2022a) and Reflexion-based reflection (Shinn et al., 2023). It takes annotated screenshots (Set-of-Mark, SoM (Yang et al., 2023a)), accessibility trees, and instructions as input, and outputs JSON-formatted actions by referencing SoM indices; (3) ASURADA, our proposed extension of M3A, which additionally incorporates real-time GPS signals and network-based context queries (e.g., weather, traffic rules, regional norms).

### 5.2 IMPLEMENTATION DETAILS

All agents operate in a common cycle consisting of interpretation, planning, execution, and reflection. While T3A relies solely on textual input, M3A grounds actions in screen content. ASURADA further integrates geographic signals to enhance reasoning and decision-making under dynamic automotive conditions.

| Base Model | Method | Input | General | | Safety-Aware | |
|---|---|---|---|---|---|---|
| | | | Explicit Control | Implicit Intent | Driving Align. | Env. Alerts |
| N/A | Human | screen | 90.0 | 82.0 | 100.0 | 88.0 |
| GPT-4o-Mini | T3A | a11y tree | 43.1 | 5.2 | 45.0 | 55 |
| | M3A | a11y tree + Screen | 52.1 | 13.6 | 50.0 | 55 |
| | ASURADA | a11y tree + Screen + GPS | 52.3 | 14.2 | 60.0 | 65 |
| Gemini 1.5 Pro | T3A | a11y tree | 30.0 | 38.0 | 55.0 | 80.0 |
| | M3A | a11y tree + Screen | 40.0 | 33.3 | 55.0 | 80.0 |
| | ASURADA | a11y tree + Screen + GPS | 43.3 | 33.3 | 90.0 | 90.0 |
| Gemini 1.5 Flash | T3A | a11y tree | 43.3 | 28.5 | 55.0 | 45.0 |
| | M3A | a11y tree + Screen | 46.6 | 33.3 | 75.0 | 55.0 |
| | ASURADA | a11y tree + Screen + GPS | 46.6 | 33.3 | 90.0 | 85.0 |
| Gemini 2.0 Flash | T3A | a11y tree | 30.0 | 38.1 | 65.0 | 55.0 |
| | M3A | a11y tree + Screen | 43.3 | 38.0 | 65.0 | 45.0 |
| | ASURADA | a11y tree + Screen + GPS | 46.6 | 45.7 | 90.0 | 70.0 |
| Gemini 2.0 Flash-Lite | T3A | a11y tree | 43.3 | 32.8 | 50.0 | 60.0 |
| | M3A | a11y tree + Screen | 33.3 | 33.3 | 42.8 | 60.0 |
| | ASURADA | a11y tree + Screen + GPS | 40.0 | 35.0 | 90.0 | 60.0 |

Table 3: Success rates (SR %) of different agent configurations on Automotive-ENV. Results are reported across *General* tasks (Explicit Control, Implicit Intent) and *Safety-Aware* tasks (Driving Alignment, Environment Alerts).

## 5.3 MAIN RESULTS

Table 3 reports the success rates of different base models under three proxy settings (T3A, M3A, ASURADA) across four task categories. Visual grounding substantially improves general task performance when T3A is enhanced with M3A (adding screen pixels). Most models show clear improvements in Explicit Control (EC) and moderate gains in Implicit Intent (II). For example, GPT-4o-Mini improves from 43.1 to 52.1 on EC and 5.2 to 13.6 on II, while Gemini 1.5 Flash advances from 43.3 to 46.6 on EC and 28.5 to 33.3 on II. However, substantial gaps remain compared to human performance (90.0 for EC, 82.0 for II), particularly in translating ambiguous goals into multi-step action plans.

Geographic context in ASURADA dramatically enhances safety performance, with Gemini models reaching 90% accuracy on Driving Alignment (DA). Gemini 1.5 Flash jumps from 55.0 to 90.0 on DA and 45.0 to 85.0 on Environment Alerts (EA), demonstrating that incorporating local priors (speed limits, road conditions) effectively reduces safety errors. Despite these improvements, models still fall short of human benchmarks (100.0 for DA, 88.0 for EA), indicating further progress is needed in safety-critical decision making.

| Dimension | Without Geo-Context | With Geo-Context |
|---|---|---|
| Input | Current speed: 80 km/h; | Current speed: 80 km/h; GPS: (48.8566, 2.3522) Location: Paris city center. Local rules snapshot: urban roads limited to 50 km/h unless otherwise posted. |
| Planning | Knowing only that the vehicle is traveling at 80 km/h, and lacking information about road type or local limits, the speed is not particularly high and should be treated as reasonably safe by default. | In Paris urban roads, where the legal limit is 50 km/h, traveling at 80 km/h is clearly above the posted limit and must be judged as unsafe and unlawful. |
| Action Decision | Takes no action, status remains infeasible. | Opens the safety notification center. (click index 22) |
| Feedback | **Fail** | **Succ** |

Table 4: Comparison of driving decisions with and without geographic context in an urban scenario. Both image information and the ally tree are also provided as inputs; this table isolates the effect of contextual differences in vehicle-related driving information.

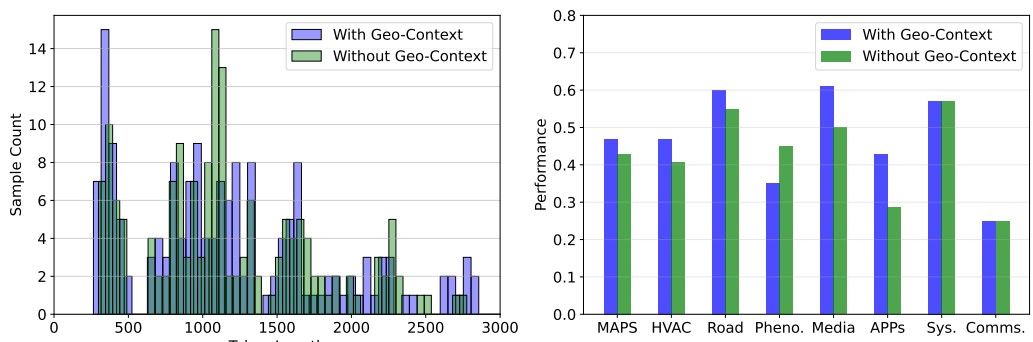

Figure 4: Comparison of inference tokens with and without GPS information. Left: distribution of token lengths. Right: task-wise performance across hotspot categories.

## 5.4 Analysis of Geo-Context

To investigate the impact of geographic context on agent decision-making, we present both qualitative and quantitative analyses. The qualitative analysis illustrates how contextual grounding alters safety judgments in representative driving scenarios, while the quantitative analysis examines its influence on reasoning efficiency and task-level performance across diverse automotive GUI tasks.

**Qualitative Analysis** Table 4 demonstrates how geographic context is essential for accurate safety alignment in driving scenarios. When only the speed of 80 km/h is given, without information about location or applicable limits, the model defaults to treating the situation as reasonably safe, since 80 km/h is not inherently excessive in many contexts. However, once geographic context is introduced—indicating that the vehicle is in central Paris, where the legal limit is 50 km/h—the same speed is recognized as both unlawful and unsafe. This shift highlights that geographic context is not only useful for refining legality judgments but also critical for preventing the model from underestimating risk. Without such contextual information, the system may overlook clear violations; with geographic grounding, it can correctly flag unsafe behavior and issue appropriate safety alerts.

**Quantitative Analysis** As shown in Figure 4, we compared token length and task performance with and without geo context. The results demonstrate that agents equipped with geo-context generate substantially more efficient reasoning trajectories: their sequences are shorter and more concentrated, with most remaining under 1500 tokens and rarely exceeding 2000. In contrast, context-blind baselines frequently produce longer outputs, with peaks in the 1000–1500 range and heavy tails approaching 3000 tokens. This indicates that environmental constraints do not complicate the reasoning process but instead reduce redundancy and improve efficiency. In terms of task performance, the impact of geo context varies across categories. Substantial gains of 15–30% are observed in tasks such as Media, MAPS, HVAC, and Road, where external environmental signals effectively disambiguate user intent and system requirements. By contrast, routine control tasks such as System and Communication show only modest improvements of 2–5%, suggesting that geo context plays a limited role in deterministic operations.

## 6 Conclusion

In this work, we present Automotive-ENV, the first large-scale benchmark explicitly designed for evaluating multimodal agents in realistic automotive GUI environments. Unlike desktop or mobile benchmarks, Automotive-ENV provides structured, reproducible, and geographically parameterized tasks that capture the complexity of in-vehicle interaction under real-world constraints. Building on this foundation, we propose ASURADA, a geo-adaptive agent capable of integrating GPS location and contextual signals to deliver safe and personalized actions. Our experiments show that geo-context integration not only improves task accuracy, especially in safety-critical settings, but also reduces reasoning overhead by enabling proactive, context-driven planning. Together, Automotive-ENV and ASURADA establish a foundation for the next generation of in-vehicle assistants that are multimodal, safety-aware, and culturally adaptive, advancing the reliable deployment of autonomous agents in high-stakes driving environments.

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
