## A    USE OF LARGE LANGUAGE MODELS

We used large language models (e.g., vision–language variants of GPT-family and Gemini-family) as assistive tools during the preparation of this paper. They were employed for language polishing, copy editing, and suggesting alternative phrasings to improve readability. LLMs were *not* used to generate research ideas, design experiments, produce results, or draw conclusions. All research contributions—including benchmark design, environment implementation, experiments, analysis, and interpretation—were performed by the authors, who take full responsibility for the content.

## B    ETHICS STATEMENT

This work does not involve human subjects, personal data, or sensitive user information. All experiments are conducted in a simulated in-vehicle GUI environment (Automotive-ENV) with programmatic task checkers. When using publicly available UI assets or icons, we respect their original licenses; any third-party materials included are either under compatible licenses or recreated by the authors. The benchmark includes safety-aware tasks (e.g., driving alignment and environment alerts); these are designed for research on safer interaction policies and must *not* be deployed to bypass driving regulations or distract drivers. Our release will exclude harmful content and will be provided strictly for research purposes under an academic license.

## C    REPRODUCIBILITY STATEMENT

We commit to releasing all artifacts required to reproduce our results: (1) the Automotive-ENV simulator, task definitions, and deterministic evaluation scripts; (2) agent baselines with configuration files, prompts, and inference parameters (e.g., temperature, max tokens); (3) logs of trajectories, screenshots, and a11y trees for completed runs; and (4) environment specifications (Python/Android tooling versions) and seed settings. We will also provide instructions and scripts for end-to-end replication—from environment setup to batch evaluation—together with hardware/OS details and expected wall-clock ranges for each experiment.

## D    DISCUSSION

GPS signals are indispensable for providing geographic context in automotive agents, yet they are prone to disruptions in real-world environments such as tunnels, underground parking, or dense urban canyons. These interruptions can cause temporary localization failures, directly undermining navigation and geo-dependent decision-making. To address this limitation, large language models (LLMs) can act as virtual sensors by leveraging their built-in knowledge of road networks together with the last available GPS coordinates and timestamps. During short signal outages, the agent can simulate intermediate positions and continue offering navigation or context-aware recommendations. Once connectivity is restored, the simulated trajectory can be aligned with actual positioning data. This capability highlights the potential of LLMs to complement imperfect sensor signals and enhance robustness in safety-critical automotive applications.