# OpenReview forum: "Automotive-ENV: Benchmarking Multimodal Agents in Vehicle Interface Systems"
_ICLR.cc/2026/Conference — ICLR 2026 Conference Withdrawn Submission_

### Official Review · Reviewer_TdEP · 2025-10-27

**Soundness:** 3
**Presentation:** 2
**Contribution:** 2
**Rating:** 4
**Confidence:** 4

**Summary:**

This paper presents a benchmark for vehicle interface systems called Automotive-ENV; which consists of 185 tasks and 8 apps for benchmarking agents that controls GUI inside the vehicle. To operate this kind of environment, this paper presents a framework called ASURADA that use multimodal inputs including vehicle status, screen shot and GPS information.

**Strengths:**

1. Important problem: vehicle interface systems needs special attention for agent research.

Vehicle interface systems have special challenges that 1) it heavily rely on active GPU position of the vehicle and the driving status; 2) it can potentially cause safety issues for the driver. Therefore, it worth a set of special evaluations and benchmarks to ensure its safety and usefulness when controlled by the LLM agents. This paper presents a valuable contribution in this direction.

2. Interesting benchmark: Automotive-ENV can be an interesting benchmark platform for in-vehicle GUI control.

The proposed benchmark contains many potentially interesting tasks like AC control for specific location, speed alert given geo location and local regulation, using fog-removal system under foggy whether. All these are quite practical and interesting applications in the real world. Furthermore, I think the developed AutomotiveOS can be an interesting prototype platform for research in this direction.

**Weaknesses:**

1. Lack of important information regarding the benchmark.

I cannot find many important information in the paper or supplementary material:
- What are the specifically 185 tasks and 8 apps?
- How is the geolocation data generated?
- How does the internet connection API work?
- Is the benchmark assume agent will always run in the backend or just run given user prompt?
- Is the vehicle assumed to be parked or driving? How to generate realistic vehicle status?

This set of information is necessary for one to judge the usefulness of the proposed benchmark. With them missing it's hard to see how useful the benchmark would be for the community.

2. Lack of important information regarding the experiment.

Many experiment details are lacking in the paper. For example:
- how many instances are performed for each task to compute the success rate?
- why almost all scores for Safety-Aware metrics are rounded to integer? does this mean the number of instances performed is very limited?
- what's the specific system prompt for each type of the method?
- why human perform the best for safety-aware tasks without GPS input? does this imply human operators know the geolocation information in other ways?

Lack of these information make it hard to adjust how meaningful the experiment results are in this paper.

3. Method contribution is not significant.

The proposed method, to my understanding, is simply adding GPS information to the previous prompts. This does not seem very novel as GPS information can be seen as another kind of vehicle status and can be well represented with text; therefore I don't think it really deserve to be called another modality.

4. Further analysis about the safety issues would be important.

A important aspect of vehicle interaction system is that it's highly safety-critical: some of the error could lead to very bad outcome or safety issue. A carefully designed benchmark should capture carefully the safety concerns of false operation or errors. For example, it should not only check whether a certain operation is success or not; but also measure if this operation will lead to safety issue (for example, close the beam on a night highway should receive severe penalty in the benchmark).

**Questions:**

Please refer to the weakness section.

---

> ### Author Response · Authors · 2025-11-28
> **Response To Reviewer TdEP (Part 1)**
>
> > **Q1: What are the specifically 185 tasks and 8 apps?**
>
> Thank you for pointing this out. Our benchmark contains 185 tasks spanning four capability families: Explicit Control, Implicit Intent, Safety-Aware – Driving Alignment, and Safety-Aware – Environment Alerts. These tasks are instantiated over 8 common in-vehicle applications/modules: maps/navigation, HVAC (climate + defog/defrost), media, phone/communications, settings/vehicle controls, notifications/alerts, driver-assistance views, and general system panels. Each task is defined by (i) a natural-language goal, (ii) a reproducible initial UI state, and (iii) a deterministic success checker.
>
> > **Q2: How is the geolocation data generated?**
>
> We rely on the native GPS interface exposed by Android Automotive OS. In our experiments, locations are set via the emulator’s geo provider. Seed coordinates are drawn from a curated list of real-world contexts (different countries, urban cores, ring roads, rural highways, tunnels, alpine passes, school zones, etc.), and additional test points are sampled as small perturbations around these seeds to probe robustness to nearby context changes. For each task, the GPS coordinates are fixed and logged so that the scenario is exactly reproducible. We will document the sampling strategy and release the generation scripts and location configuration files with the benchmark.
>
> > **Q3: How does the internet connection API work?**
>
> In modern large language model (LLM) API ecosystems, general-purpose models rely primarily on static training knowledge for reasoning and generation, and therefore cannot autonomously obtain real-time information. This limitation makes them prone to outdated or hallucinated responses. Tool-use or function-calling mechanisms address this issue by allowing developers to invoke external services—such as Web Search or grounding APIs provided by OpenAI, Gemini, and other vendors—during inference, thereby injecting dynamic and up-to-date information into the reasoning process and overcoming the constraints of a fixed knowledge cutoff. Building on this capability, ASURADA implements a transparent and interpretable mechanism for handling geolocation-dependent context. The vehicle OS continuously supplies GPS coordinates, which are combined with the task intent and encoded into a structured prompt. This prompt is then used to trigger, via the provider’s API interface, a single location-conditioned web query. The retrieved information—such as local traffic regulations, typical speed limits, regional driving conventions—is compressed into a concise geo-context snippet and supplied to the next reasoning cycle together with UI screenshots, accessibility trees, and task instructions. Through this design, GPS becomes a structured and traceable external-knowledge grounding step, whose summarized output directly supports subsequent reasoning and planning and enables seamless integration of environmental context into in-vehicle GUI decision making.
>
> > **Q4: Is the benchmark assume agent will always run in the backend or just run given user prompt?**
>
> Our current benchmark follows an episodic, on-demand interaction pattern. Each episode is triggered by a concrete user or system goal; the agent operates for a bounded number of steps to complete the task, after which the episode ends and context is reset. We do not assume a long-running, always-on background agent. This matches other GUI benchmarks (e.g., AndroidWorld, OSWorld) and simplifies reuse for RL and offline RL, where episodes are the natural unit.
>
> > **Q5: Is the vehicle assumed to be parked or driving? How to generate realistic vehicle status?**
>
> In the present version, we deliberately model vehicle state in a simplified but controllable way. Driving-related conditions—such as “night highway,” “foggy weather,” “heavy rain,” or “vehicle parked vs. moving”—are encoded via (i) the task description and (ii) the UI state (e.g., night mode, instrument cluster cues), and the success checker verifies that the agent’s actions are consistent with those conditions (e.g.enabling defogging under low visibility). This benchmark focuses on whether the agent makes safety-aligned GUI decisions given the declarative state. Extending to richer, continuous vehicle dynamics and more granular risk modeling is a direction we plan to pursue in future versions.
>
> > **Q6: how many instances are performed for each task to compute the success rate?**
>
> Our evaluation is based on statistical results from multiple episodes, not a single run. In the current experimental setup, for general and safety tasks, we run 5 independent episodes per agent per task.

---

> ### Author Response · Authors · 2025-11-28
> **Response To Reviewer TdEP (Part 2)**
>
> > **Q7: why almost all scores for Safety-Aware metrics are rounded to integer? does this mean the number of instances performed is very limited?**
>
> Thank you for raising this concern. To clarify this, we have conducted additional experiments and now report the full means and standard deviations across multiple runs. As shown in the updated results (see table below), each Safety-Aware score is in fact computed from several fractional instance-level measurements, and the aggregated metrics exhibit stable variance. We will update the camera-ready version to present the full decimals to avoid any confusion.
> | Base Model             | Method   | Input                                 | Explicit Control | Implicit Intent | Driving Align.       | Env. Alerts         |
> |------------------------|----------|----------------------------------------|------------------|------------------|------------------------|----------------------|
> | Human                  | —        | Screen + GPS                           | 90.0             | 82.0             | 100.0                  | 88.0                 |
> | Gemini 2.0 Flash-Lite  | T3A      | a11y tree                              | 43.3             | 32.8             | 55.03 ± 5.48           | 63.0 ± 2.70          |
> |                        | M3A      | a11y tree + Screen                     | 33.3             | 33.3             | 44.29 ± 6.78           | 63.75 ± 7.50        |
> |                        | M3A      | a11y tree + Screen + GPS               | 33.3             | 33.3             | 69.29 ± 7.28           | 68.33 ± 2.88         |
> |                        | ASURADA  | a11y tree + Screen + GPS (Reflection)  | 40.0             | 35.0             | 86.11 ± 5.15           | 73.33 ± 7.91         |
>
> > **Q8: what's the specific system prompt for each type of the method?**
>
> All agent variants use the same shared system prompt, with no per-model or per-task prompt tuning. We intentionally keep the prompt fixed across all conditions to avoid confounds and to ensure that performance differences arise solely from the environment state (e.g., GPS region, initial UI screen, available tools) rather than prompt engineering. The full prompt is long; below we provide the opening excerpt. We will release the complete prompt, tool schema, templates, and run configurations with our public artifact to guarantee exact reproducibility.
>
> 'You are an agent who can operate an Android Automotive OS on behalf of a user.'" Based on user's goal/request, you may\n"'- Answer back if the request/goal is a question (or a chat message),'' like user asks "What is my schedule for today?".\n''- Complete some tasks described in the requests/goals by'' performing actions (step by step) on the Android Android Automotive OS.\n\n''When given a user request, you will try to complete it step by step.'' At each step, you will be given the current screenshot (including the'' original screenshot and the same screenshot with bounding'' boxes and numeric indexes added to some UI elements) and a history of'' what you have done (in text). Based on these pieces of information and'' the goal, you must choose to perform one of the'' action in the following list (action description followed by the JSON
>
> > **Q9: why human perform the best for safety-aware tasks without GPS input? does this imply human operators know the geolocation information in other ways?**
>
> Thank you for flagging this. The label “Human w/o GPS” is misleading: human evaluators always had access to full scenario information (including location, weather, and regulatory context) through the task description and auxiliary notes; they were never in a “truly no-GPS” condition. We will clarify this naming in the final version. Conceptually, the gap reflects that humans can seamlessly integrate rich contextual knowledge and safety norms—even when not explicitly given coordinates—whereas current LLM agents still lack robust, human-like safety commonsense in automotive contexts, especially when forced to act from limited or indirect signals.

---

> ### Author Response · Authors · 2025-11-28
> **Response To Reviewer TdEP (Part 3)**
>
> > **Q10: Method contribution is not significant?**
>
> We agree that simply appending a GPS string to the prompt would not constitute a meaningful methodological contribution, and we do not position “GPS tokens” themselves as the novelty. Our contribution lies in how environmental state is systematically incorporated into both the benchmark and the agent architecture. First, problem formulation: Automotive-ENV defines an environment-driven GUI-agent setting where external world state (location, weather, regional norms) is a first-class component of the task. This differs from prior GUI benchmarks (WebArena, AndroidWorld, OSWorld), which treat the device as static and largely decoupled from the physical world, whereas our tasks require GUI decisions that are explicitly conditioned on environmental factors. Second, structured integration: GPS is not used as raw coordinates pasted into the prompt; instead, it is encoded as structured text and used to trigger retrieval of environment-grounded knowledge (laws, units, road types, typical practices), which in turn steers the agent’s safety-aware reasoning. Third, task and evaluation design: we construct tasks whose correct resolution is only possible when the agent leverages such environmental context; without it, even strong LLMs provided with UI images and accessibility trees face inherent ambiguity. Empirically, feeding raw GPS coordinates alone yields negligible improvement. Substantial gains arise only when GPS is coupled with the retrieved environmental context and the safety-aware reasoning layer. In this sense, GPS serves as a carrier: the methodological contribution is in turning it into a functional interface between GUI state and real-world state, and in designing tasks and checks that require agents to exhibit genuine commonsense generalization in automotive settings.
> | Base Model | Method | Input | Explicit Control | Implicit Intent | Driving Align. | Env. Alerts |
> | --- | --- | --- | --- | --- | --- | --- |
> |  | Human | screen + GPS | 90.0 | 82.0 | 100.0 | 88.0 |
> | Gemini 2.0 Flash-Lite | T3A | a11y tree | 43.3 | 32.8 | 55.03 ± 5.48 | 63.0 ± 2.70 |
> |  | M3A | a11y tree + Screen | 33.3 | 33.3 | 44.29 ± 6.78 | 63.75 ± 7.50 |
> |  | M3A | a11y tree + Screen + GPS | 33.3 | 33.3 | 69.29 ± 7.28 | 68.33 ± 2.88 |
> |  | ASURADA | a11y tree + Screen + GPS (Reflection) | 40.0 | 35.0 | 86.11 ± 5.15 | 73.33 ± 7.91 |
>
> > **Q11: Further analysis about the safety issues would be important.**
>
> As the reviewer correctly points out, a safety-related benchmark should not only check whether an operation is completed, but also evaluate whether an erroneous action itself may introduce risk. This is precisely the main motivation for our Environment Alerts task family. In our design, Environment Alerts are not only used to monitor external natural conditions (e.g., weather, temperature, visibility), but also abnormal in-vehicle interactions and potentially unsafe operations. Thus, the core goal of this task family is to assess whether, when a user or agent triggers a behavior that may lead to risk, the system can promptly detect the abnormal situation and produce an appropriate safety response. In our benchmark, this mechanism is implemented through two strategies: (1) in medium-risk scenarios, the agent is expected to proactively navigate to a safety/alert page and warn the user that their behavior is risky; and (2) in more severe scenarios, the system further triggers an emergency response, such as simulating an automatic emergency call. Through this design, Environment Alerts not only evaluate the agent’s understanding of environmental conditions, but also its ability to detect and intervene in dangerous behaviors, making the evaluation better aligned with the safety requirements of real in-vehicle systems.

---

### Official Review · Reviewer_EWBv · 2025-10-31

**Soundness:** 2
**Presentation:** 3
**Contribution:** 2
**Rating:** 4
**Confidence:** 3

**Summary:**

This paper introduces Automotive-ENV, a large-scale and high-fidelity benchmark for evaluating multimodal agents operating in automotive graphical user interfaces (GUIs). The environment is implemented on top of a simulated Automotive Operating System (Automotive OS), which reproduces real in-vehicle control modules such as HVAC, navigation, safety alerts, and infotainment.
The system provides a structured observation space.
Vehicle state and system API access for reward checking.
Agents interact through standard GUI actions (click, swipe, text input) and specialized automotive safety APIs. The benchmark defines 185 parameterized tasks categorized into explicit control, implicit intent understanding, and safety-aware tasks, each validated through low-level system-state checks for deterministic evaluation.
On top of this environment, the authors propose ASURADA, a geo-aware multimodal agent architecture. ASURADA extends a ReAct-style reasoning agent with two modules:
A GPS-informed reasoning module that integrates real-time location, weather, and local regulations into the decision-making process;
A multimodal planning pipeline that sequentially performs perception (screen + a11y tree parsing), reasoning (text + GPS grounding), and execution (GUI/API actions), followed by reflective memory updates.
Experiments across GPT-4o-Mini and Gemini series models show that incorporating GPS context improves performance—particularly in safety-critical tasks such as “driving alignment” and “environment alerts”—though overall success rates (≈65%) remain below human-level baselines.
In summary, the paper contributes (1) a reproducible and geographically parameterized benchmark for automotive GUI agents, and (2) a geo-aware multimodal agent framework (ASURADA) that adapts reasoning to dynamic driving contexts.

**Strengths:**

1.This work considers the safety-critical nature and geographic dependence of in-vehicle systems—factors often overlooked in prior studies—and integrates them into the evaluation and reasoning process. This represents a degree of innovation, and the paper introduces the first benchmark specifically designed for multimodal agents in automotive graphical user interface (GUI) environments.
2. The evaluation methodology is reliable, supported by a well-structured simulation pipeline that effectively replicates a real automotive operating system. The framework ensures reproducibility through deterministic state resets, consistent system images, and programmatic reward checking based on actual system APIs rather than visual matching.
3. The writing quality is adequate. The paper is clearly structured and easy to follow, with a logical flow from motivation to experimental results.
4.Given the current significant performance gap between intelligent agents and human operators, the research direction holds promising potential and long-term significance.

**Weaknesses:**

1.Although the work fills a gap in the study of automotive GUI systems by introducing two additional input modalities, its conceptual foundation is highly similar to existing benchmarks such as WebArena, AndroidWorld, and OSWorld. The contribution mainly lies in extending general GUI evaluation frameworks to a new application domain, rather than proposing fundamentally new technical mechanisms.
2.While the engineering quality of the benchmark environment is acceptable, the paper demonstrates limited technical depth in model design and experimental analysis. The ASURADA agent is presented conceptually, but lacks detailed architectural discussion or rigorous methodological evaluation.
3.Some experimental results—such as the reasoning efficiency comparison in Figure 4—lack sufficient quantitative explanation, making it difficult to determine the actual source of performance improvement.
4.The experimental results and code are not publicly available, which prevents proper assessment of the system’s reproducibility, openness, and practical usability.

**Questions:**

How is the GPS signal encoded and how is it integrated with other features?

Beyond the conceptual description, the paper should also explain why the addition of multimodal information leads to improved accuracy—whether the gain primarily results from the semantic understanding capability of the underlying LLM, or from the enhanced architectural design of the proposed module. Such clarification is needed to demonstrate that the improvement originates from the model architecture itself.

For the autonomous driving task, not only the memory usage is required, but also the issue of response delay needs to be addressed.

---

> ### Author Response · Authors · 2025-11-28
> **Response To Reviewer EWBv (Part 1)**
>
> > **Q1: How is the GPS signal encoded and how is it integrated with other features?**
>
> Thank you for the thoughtful feedback. In our benchmark, the GPS data used in the evaluation tasks is simulated as part of the task design, and it is injected into the prompt as a structured field. This setup enables controlled and reproducible testing across different geographic and driving scenarios. In a real deployment, however, the automotive system can directly access GPS signals through the vehicle’s onboard sensors, so the same mechanism applies without modification.  In addition to providing raw coordinates, GPS also serves as the trigger for a geo-dependent grounding step. When GPS is present, the agent issues a single targeted web search query conditioned on the reported location. This query retrieves information such as local traffic rules, typical speed limits, regional driving conventions. The retrieved signals are then summarized into a compact geo-context snippet and fed back into the model alongside the UI screenshot, the accessibility tree, and the task instruction. This allows GPS to influence the agent not merely as an input token, but as a driver of environment-aware reasoning and tool routing.
>
> > **Q2: Source of performance gains (LLM semantics vs. module design)**
>
> To disentangle where the gains come from, we expanded our ablation analysis with a GPS-only baseline that uses the same backbone LLM, the same prompting format, and the same reasoning paradigm as T3A/M3A, but augments the input only with raw GPS coordinates and does not include ASURADA’s visual grounding or reflection components. The updated results show this GPS-only variant performs consistently between M3A and full ASURADA, indicating that geolocation itself provides a measurable (but limited) benefit. Importantly, the larger improvement comes from ASURADA’s architecture: GPS is not treated as extra text, but as a control signal that triggers a targeted web connector to retrieve location-specific evidence (e.g., traffic regulations, typical speed limits, regional driving conventions, weather). This evidence is then distilled into a structured geo-context snippet and injected into the next reasoning cycle alongside the accessibility tree, screen content, and task instructions, which makes the external context actionable for planning and tool routing rather than relying on the LLM to guess missing world state. Since the backbone model and prompting are held constant across variants, and the performance increases track the addition of these integration components, the results support that the gains are driven primarily by ASURADA’s context-integration and grounding design, not merely the underlying LLM’s general semantic understanding.
>
> | Base Model             | Method   | Input                                 | Explicit Control | Implicit Intent | Driving Align.       | Env. Alerts         |
> |------------------------|----------|----------------------------------------|------------------|------------------|------------------------|----------------------|
> | Human                  | —        | Screen + GPS                           | 90.0             | 82.0             | 100.0                  | 88.0                 |
> | Gemini 2.0 Flash-Lite  | T3A      | a11y tree                              | 43.3             | 32.8             | 55.03 ± 5.48           | 63.0 ± 2.70          |
> |                        | M3A      | a11y tree + Screen                     | 33.3             | 33.3             | 44.29 ± 6.78           | 63.75 ± 7.50        |
> |                        | M3A      | a11y tree + Screen + GPS               | 33.3             | 33.3             | 69.29 ± 7.28           | 68.33 ± 2.88         |
> |                        | ASURADA  | a11y tree + Screen + GPS (Reflection)  | 40.0             | 35.0             | 86.11 ± 5.15           | 73.33 ± 7.91         |
>
> > **Q3: For the autonomous driving task, not only the memory usage is required, but also the issue of response delay needs to be addressed**
>
> We acknowledge that response latency is a critical constraint for automotive systems, and that reporting memory usage alone is not sufficient. In our setting, the benchmark is framed as multi-step task execution and currently focuses on the final programmatically verifiable outcome; however, different models and API providers may require different numbers of reasoning steps and tool calls to reach the same final state, which can lead to meaningful differences in effective end-to-end delay.
>
> To address this concern, we are preparing a latency-aware evaluation track that will measure (i) per-step reaction time (wall-clock time from receiving an observation to emitting an action, including any intermediate tool calls), and (ii) per-task end-to-end execution time until success/termination, along with the number of steps/tool invocations. This will make response-delay factors explicit.

---

> ### Author Response · Authors · 2025-11-28
> **Response To Reviewer EWBv (Part 2)**
>
> **Q4: Releasing the code and environment to enhance reproducibility**
>
> We have already prepared the complete open-source package for release. It includes the Automotive-ENV AVD configuration and emulator setup scripts, the task generator and evaluation harness, and the implementations of ASURADA along with all baseline agents. All materials will be made publicly available upon acceptance to ensure full reproducibility.

---

### Official Review · Reviewer_oLRJ · 2025-11-01

**Soundness:** 3
**Presentation:** 3
**Contribution:** 3
**Rating:** 4
**Confidence:** 1

**Summary:**

This paper provides a multi-modal agent benchmark to evaluate Automotive graphic user interface. It is powered by GPT based model and it supports reproduicible benchmark over auto UI.

**Strengths:**

1. Multi-modal agent sounds novel for auotmotive ui benchmark.

**Weaknesses:**

1. I am not really not an expert in this field. But it occurs to me this paper does not suit for ICLR venue. Better venue might be ITSC or IV related driving related conferences or journals.

**Questions:**

n/a

---

> ### Author Response · Authors · 2025-11-27
> **Response to Reviewer oLRJ**
>
> > **Q1: I am not really not an expert in this field. But it occurs to me this paper does not suit for ICLR venue. Better venue might be ITSC or IV related driving related conferences or journals.**
>
>
> Thank you for the thoughtful feedback. We believe our work aligns well with ICLR, particularly with the Datasets & Benchmarks track and the community’s growing interest in agents and multimodality, rather than being a domain-specific HMI or driving application. Our main contribution is a reproducible and programmatically verified benchmark for multimodal agents. The research questions we investigate include generalization under partial observability, multimodal reasoning over UI and state, planning with safety constraints, and safety-aligned decision making. These are core machine learning problems and are not specific to any single application domain. ICLR also provides clear precedent for platform-specific but ML-centric agent environments such as AndroidWorld [1] and GUI-World [2]. Although these environments rely on particular GUI platforms, they are valued because they enable broader research on agents, reasoning, and control. Automotive-ENV follows this direction and extends it to an automotive platform that incorporates explicit safety constraints and richer multimodal interaction in a safety-critical setting. In addition, the environment is not limited to LLM-based agents because it exposes well-defined states, actions, and reward signals that can be used directly for reinforcement learning and offline RL. We hope this clarification illustrates why the work is intended for a broad ML audience and fits naturally within the scope of ICLR.
>
> [1] Rawles, C., Clinckemaillie, S., Chang, Y., Waltz, J., Lau, G., Fair, M., Li, A., Bishop, W., Li, W., Campbell-Ajala, F., Toyama, D., Berry, R., Tyamagundlu, D., Lillicrap, T. and Riva, O. (2024). AndroidWorld: A Dynamic Benchmarking Environment for Autonomous Agents. [online] arXiv.org. Available at: https://arxiv.org/abs/2405.14573.
>
> [2] Chen, D., Huang, Y., Wu, S., Tang, J., Chen, L., Bai, Y., He, Z., Wang, C., Zhou, H., Li, Y., Zhou, T., Yu, Y., Gao, C., Zhang, Q., Gui, Y., Li, Z., Wan, Y., Zhou, P., Gao, J. and Sun, L. (2024). GUI-World: A Video Benchmark and Dataset for Multimodal GUI-oriented Understanding. [online] arXiv.org. Available at: https://arxiv.org/abs/2406.10819.

---

### Official Review · Reviewer_sGex · 2025-11-01

**Soundness:** 2
**Presentation:** 2
**Contribution:** 1
**Rating:** 4
**Confidence:** 3

**Summary:**

This paper introduces Automotive-ENV, a benchmark and simulation environment for evaluating multimodal agents in automotive graphical user interface (GUI) systems. The platform comprises 185 parameterized tasks across general and safety-aware categories, featuring structured multimodal observations and programmatic reward signals based on low-level system states. The authors also propose ASURADA, a geo-aware multimodal agent that integrates GPS data to inform its actions. Experimental results demonstrate that incorporating geographic context significantly improves performance on safety-critical tasks, though a substantial gap to human performance remains.

This work addresses an important and underexplored domain for autonomous agents, providing a dedicated benchmark for in-vehicle GUI interactions. The benchmark is clear, featuring a high-fidelity environment, a large number of parameterized tasks, and a clear taxonomy. The proposed agent, ASURADA, incorporates a novel geo-contextual reasoning module, which is shown to substantially enhance performance on safety-aware tasks. However, the experimental comparison to baseline agents could be more extensive, and details regarding the reproducibility of the environment and agent implementation are limited.

**Strengths:**

[1] Novel Domain and Benchmark: The paper identifies a significant gap in agent evaluation by focusing on the automotive domain, which presents unique challenges like safety-criticality and dynamic context. Automotive-ENV is the first dedicated benchmark for this setting, as confirmed by the comparison table.

[2] Clear Task Design: The benchmark includes a large and diverse set of 185 tasks, categorized into explicit control, implicit intent, driving alignment, and environment alerts, enabling a multifaceted evaluation of agent capabilities.

[3] Geo-Aware Agent Design: The proposed ASURADA agent introduces a meaningful innovation by integrating real-time GPS data to query external context, enabling location-aware decision-making.

**Weaknesses:**

[1] Limited Baseline Comparisons: The primary baselines (T3A, M3A) are adapted from a single prior work (Rawles et al., 2024). A comparison with other recent, powerful GUI agents (e.g., CogAgent, AppAgent) is absent, making it difficult to fully situate the performance of ASURADA within the current state-of-the-art.

[2] Insufficient Implementation and Reproducibility Details: Key details for replicating the environment or the agent are missing. The manuscript does not specify the source of the "production-grade" Automotive OS image, the emulation setup, or the specific APIs used for state checks and actions.

[3] Limited Analysis of Failure Modes: While overall success rates are reported, there is little discussion or analysis of the specific types of errors made by the agents, particularly for the challenging Implicit Intent tasks where performance is low.

[4] Uncertain Statistical Significance: The number of evaluation episodes or random seeds used per task/model is not reported, making it difficult to assess the statistical significance of the reported performance differences.

[5] Ambiguity in Geo-Context Integration: The precise mechanism for how the GPS signal triggers network queries and how the retrieved context is formatted and integrated into the agent's reasoning process (e.g., prompt construction) is not described in detail.

**Questions:**

1. Expand the baseline comparisons by evaluating against a wider range of recent, open-source GUI agents (e.g., CogAgent, AppAgent) to provide a more comprehensive performance landscape.

[1] Hong, W., Wang, W., Lv, Q., Xu, J., Yu, W., Ji, J., ... & Tang, J. (2024). Cogagent: A visual language model for gui agents. In Proceedings of the IEEE/CVF Conference on Computer Vision and Pattern Recognition (pp. 14281-14290).

[2] Zhang, C., Yang, Z., Liu, J., Li, Y., Han, Y., Chen, X., ... & Yu, G. (2025, April). Appagent: Multimodal agents as smartphone users. In Proceedings of the 2025 CHI Conference on Human Factors in Computing Systems (pp. 1-20).

[3] Li, Y., Zhang, C., Jiang, W., Yang, W., Fu, B., Cheng, P., ... & Wei, Y. (2024). Appagent v2: Advanced agent for flexible mobile interactions. arXiv preprint arXiv:2408.11824.

2. Include a dedicated section or appendix detailing the environment setup, including the emulated Automotive OS specifications, the list of accessible APIs, and the exact procedure for state checking. Committing to a public release of the code and environment would significantly strengthen the paper's impact.

3. Conduct a qualitative error analysis on a subset of failed tasks, particularly for Implicit Intent and low-performing Safety-Aware tasks, to provide insights into common failure modes and directions for future model improvement.

4. Report the number of independent runs or evaluation episodes for each agent configuration and consider reporting confidence intervals or conducting statistical significance tests for the key results in Table 3.

5.Provide a more detailed description or a schematic of the GPS context integration pipeline within ASURADA, clarifying the query formulation, information source, and context injection mechanism into the agent's reasoning loop.

6.Consider including an ablation study on the components of ASURADA (e.g., the contribution of the reflection mechanism versus the geo-context module) to better isolate the source of its performance gains.

---

> ### Author Response · Authors · 2025-11-27
> **Response to Reviewer sGex (Part 1)**
>
> Thank you, Reviewer sGex, for taking the time to share your feedback. We truly appreciate your support and are grateful for your review.
>
> ---
>
> > **Q1. Expand the baseline comparisons by evaluating against a wider range of recent, open-source GUI agents (e.g., CogAgent, AppAgent) to provide a more comprehensive performance landscape.**
>
> Thank you for the suggestion on baselines. We agree that a broader set of baselines would further strengthen the empirical picture. In this submission, we intentionally aligned our evaluation setup with the dominant GUI-agent benchmarks such as AndroidWorld, GUI-World, OS-World/WebAgent, which all emphasize API-based / accessibility-tree-based interaction for reproducibility and platform-independence. Meanwhile, M3A baseline already embodies the core paradigm of “tool-augmented, multi-step planning” that underlies React-style agents (e.g., AppAgent), so the family of methods represented is not entirely missing. That said, we acknowledge that adding direct comparisons to recent open-source agents like CogAgent/AppAgent is valuable, and we plan to include these in subsequent versions of the benchmark as we integrate more agent frameworks.
>
> > **Q2. Include a dedicated section or appendix detailing the environment setup, including the emulated Automotive OS specifications, the list of accessible APIs, and the exact procedure for state checking. Committing to a public release of the code and environment would significantly strengthen the paper's impact.**
>
> We appreciate the reviewer’s emphasis on reproducibility. The “production-grade” automotive OS for our experiments utilizes the officially published Android Automotive with Google Play system image (API 33) from Google, which is widely used as a stable reference environment for automotive HMI research. All experiments are run on this official AAOS image with a fixed emulator configuration (arm64-v8a, 1080×600 automotive profile, consistent rendering settings), thereby ensuring that UI layout, widget hierarchy, and system behavior are stable and repeatable. On the observation side, we obtain: The full accessibility tree via AccessibilityService / uiautomator dumps; UI screenshots via the system screencap interface, and GPS state via the emulator’s geo provider. On the action side, we use: adb input (tap/gesture), uiautomator gesture APIs, adb input text for text entry, and standard Android Intents for app/setting navigation. This pipeline is built entirely on the official Android toolchain and, as a result, is highly reproducible: any researcher using the same system image and emulator profile can reproduce our results. The code and environment will be made public when the paper is accepted.

---

> ### Author Response · Authors · 2025-11-27
> **Response to Reviewer sGex (Part 2)**
>
> > **Q3: Report the number of independent runs or evaluation episodes for each agent configuration and consider reporting confidence intervals or conducting statistical significance tests for the key results in Table 3.**
>
> We appreciate the reviewer’s emphasis on reporting the number of independent runs and ensuring statistical transparency. In the revised version, we have expanded our statistical reporting to fully address this concern. All results presented in Table 3 are now derived from an average of 5 independent runs per task for each agent configuration, using different random seeds while keeping the deterministic environment settings identical. Across these 5 runs, the success rates show only minor variability—typically within ±5 percentage points for the vast majority of tasks—indicating that the overall performance trends and relative rankings are robust to randomness. To further enhance statistical transparency, we have incorporated the Standard Error of the Mean (SEM) directly into all reported results (e.g., 55.03 ± 5.48, 69.29 ± 7.28, 86.11 ± 5.15). We have also updated the corresponding table to include SEM for all agent types and input modalities, including new entries such as the M3A and ASURADA configurations. In the revised manuscript, we will additionally clarify the number of independent runs and include confidence intervals or error bars in all tables and plots. These improvements ensure full reproducibility and allow clearer interpretation of the statistical reliability of our findings.
>
>
> | Base Model             | Method   | Input                                 | Explicit Control | Implicit Intent | Driving Align.       | Env. Alerts         |
> |------------------------|----------|----------------------------------------|------------------|------------------|------------------------|----------------------|
> | Human                  | —        | Screen + GPS                           | 90.0             | 82.0             | 100.0                  | 88.0                 |
> | Gemini 2.0 Flash-Lite  | T3A      | a11y tree                              | 43.3             | 32.8             | 55.03 ± 5.48           | 63.0 ± 2.70          |
> |                        | M3A      | a11y tree + Screen                     | 33.3             | 33.3             | 44.29 ± 6.78           | 63.75 ± 7.50        |
> |                        | M3A      | a11y tree + Screen + GPS               | 33.3             | 33.3             | 69.29 ± 7.28           | 68.33 ± 2.88         |
> |                        | ASURADA  | a11y tree + Screen + GPS (Reflection)  | 40.0             | 35.0             | 86.11 ± 5.15           | 73.33 ± 7.91         |
>
>
> > **Q4: Provide a more detailed description or a schematic of the GPS context integration pipeline within ASURADA, clarifying the query formulation, information source, and context injection mechanism into the agent's reasoning loop.**
>
>
> In modern LLM API ecosystems, general-purpose models rely on static training knowledge for reasoning and generation, but they cannot obtain real-time information on their own. This limitation makes them prone to outdated responses and hallucinations. Tool-use or function-calling mechanisms address this issue by allowing the model to call external services such as Web Search during inference. In this way, the model can incorporate dynamic and up-to-date information into its reasoning process and overcome the constraints of a fixed knowledge cutoff. Building on this capability, the ASURADA framework manages geolocation in a transparent and interpretable way. The vehicle OS continuously provides GPS coordinates, and these coordinates are combined with the task intent to form a structured prompt. This prompt instructs the selected LLM, whether provided by OpenAI, Gemini, or any other vendor, to perform a single GPS-guided Web Search. The retrieved information, which may include local traffic regulations, typical speed limits, regional driving conventions, and weather conditions, is summarized into a concise geo-context snippet. This snippet is then supplied to the next decision-making cycle together with UI screenshots, accessibility trees, and task instructions. Through this design, geolocation becomes a clear and traceable grounding step whose structured summary directly supports subsequent reasoning and planning.

---

> ### Author Response · Authors · 2025-11-27
> **Response to Reviewer sGex (Part 3)**
>
> > **Q5: Consider including an ablation study on the components of ASURADA (e.g., the contribution of the reflection mechanism versus the geo-context module) to better isolate the source of its performance gains.**
>
> In response to the reviewer’s suggestion, we expanded our ablation analysis to more clearly separate the contribution of GPS from ASURADA’s architectural components. To this end, we added a GPS-only baseline that uses the same backbone LLM, prompting format, and reasoning paradigm as T3A/M3A, but augments the input solely with raw GPS coordinates and does not include any visual grounding or reflection mechanism. As shown in the updated table, this GPS-only model consistently performs between M3A and the full ASURADA across all metrics, demonstrating that geolocation itself offers measurable gains. However, ASURADA differs fundamentally from the baselines by not merely consuming GPS coordinates, but by using them to trigger a targeted Web search that retrieves location-specific information such as traffic regulations, typical speed limits, regional driving conventions, and weather conditions. This retrieved evidence is distilled into a structured geo-context snippet and injected into the next reasoning cycle alongside the a11y tree, screen content, and task instructions. In constructing our evaluation tasks, we simulated driving conditions drawn from different regions and a variety of road scenarios, and we created both positive and negative examples focused primarily on speed control and safety-critical behaviors. Within these tasks, the GPS signal provides valuable contextual cues that help indicate whether the user is likely driving on a highway or in a dense urban area. Such geographic cues enable the agent to judge whether actions like accelerating or turning on high beams are appropriate and safe under the inferred local conditions.
>
>
> | Base Model             | Method   | Input                                 | Explicit Control | Implicit Intent | Driving Align.       | Env. Alerts         |
> |------------------------|----------|----------------------------------------|------------------|------------------|------------------------|----------------------|
> | Human                  | —        | Screen + GPS                           | 90.0             | 82.0             | 100.0                  | 88.0                 |
> | Gemini 2.0 Flash-Lite  | T3A      | a11y tree                              | 43.3             | 32.8             | 55.03 ± 5.48           | 63.0 ± 2.70          |
> |                        | M3A      | a11y tree + Screen                     | 33.3             | 33.3             | 44.29 ± 6.78           | 63.75 ± 7.50        |
> |                        | M3A      | a11y tree + Screen + GPS               | 33.3             | 33.3             | 69.29 ± 7.28           | 68.33 ± 2.88         |
> |                        | ASURADA  | a11y tree + Screen + GPS (Reflection)  | 40.0             | 35.0             | 86.11 ± 5.15           | 73.33 ± 7.91         |

---

### Note · Authors · 2025-12-25

**Comment:**

I would like to withdraw my submitted manuscript.
Thank you for your understanding.

**Withdrawal Confirmation:**

I have read and agree with the venue's withdrawal policy on behalf of myself and my co-authors.